# How migratory thrushes conquered northern North America: a comparative phylogeography approach

Carrie M. Topp[1], Christin L. Pruett[2], Kevin G. McCracken[1] and Kevin Winker[1]

[1] University of Alaska Museum and Institute of Arctic Biology, University of Alaska Fairbanks, Fairbanks, AK, USA
[2] Florida Institute of Technology, Department of Biological Sciences, Melbourne, FL, USA

Corresponding author
Kevin Winker,
kevin.winker@alaska.edu

## ABSTRACT

Five species of migratory thrushes (Turdidae) occupy a transcontinental distribution across northern North America. They have largely overlapping breeding ranges, relatively similar ecological niches, and mutualistic relationships with northern woodland communities as insectivores and seed-dispersing frugivores. As an assemblage of ecologically similar species, and given other vertebrate studies, we predicted a shared pattern of genetic divergence among these species between their eastern and western populations, and also that the timing of the coalescent events might be similar and coincident with historical glacial events. To determine how these five lineages effectively established transcontinental distributions, we used mitochondrial cytochrome *b* sequences to assess genetic structure and lineage coalescence from populations on each side of the continent. Two general patterns occur. Hermit and Swainson's thrushes (*Catharus guttatus* and *C. ustulatus*) have relatively deep divergences between eastern and western phylogroups, probably reflecting shared historic vicariance. The Veery (*C. fuscescens*), Gray-cheeked Thrush (*C. minimus*), and American Robin (*Turdus migratorius*) have relatively shallow divergences between eastern and western populations. However, coalescent and approximate Bayesian computational analyses indicated that among all species as many as five transcontinental divergence events occurred. Divergence within both Hermit and Swainson's thrushes resembled the divergence between Gray-cheeked Thrushes and Veeries and probably occurred during a similar time period. Despite these species' ecological similarities, the assemblage exhibits heterogeneity at the species level in how they came to occupy transcontinental northern North America but two general continental patterns at an among-species organizational level, likely related to lineage age.

## INTRODUCTION

It has been suggested that the North American avifauna is a composite of species with different colonization and isolation histories because multiple phylogeographic patterns

are seen in many species that are presently co-distributed (*Zink, 1996*; *Avise, 2000*; *Carstens et al., 2005*). However, closely related and ecologically similar species may be more tightly associated with each other over time compared to groups of co-occurring species that are ecologically varied (*Richman & Price, 1992*; *Webb, 2000*; *Webb et al., 2002*; *Lovette & Hochachka, 2006*).

In this study we examined five migratory thrush species with breeding ranges across northern North America: Hermit Thrush (*Catharus guttatus*), Swainson's Thrush (*C. ustulatus*), Gray-cheeked Thrush (*C. minimus*), Veery (*C. fuscescens*), and American Robin (*Turdus migratorius*). These species are relatively common members of northern woodland bird assemblages, and their breeding ranges are mostly or partly overlapping (Fig. 1; *Jones & Donovan, 1996*; *AOU, 1998*; *Sallabanks & James, 1999*; *Mack & Yong, 2000*; *Lowther et al., 2001*; *Maskoff, 2005*). They are each others' closest relatives in these communities, excluding Bicknell's Thrush (*C. bicknelli*), which we did not include because it has a small breeding range only on the eastern side of the continent (*AOU, 1998*; *Rimmer et al., 2001*; *Klicka, Voelker & Spellman, 2004*; *Winker & Pruett, 2006*; *Voelker, Bowie & Klicka, 2013*). We also excluded two other thrush species whose ranges do not span the continent: Varied Thrush (*Zoothera naevia*) and Wood Thrush (*Hylocichla mustelina*; *AOU, 1998*).

The five species chosen for this study occur in a variety of woodlands and occupy—on a community scale—similar niches as forest and woodland mutualists (all of them are insectivores, seasonal frugivores, and seed dispersers); they are likely to be each others' closest competitors in these communities (*Bent, 1949*; *Jones & Donovan, 1996*; *Sallabanks & James, 1999*; *Mack & Yong, 2000*; *Lowther et al., 2001*; *Maskoff, 2005*). Population genetics and phylogeography at the continental scale, paired with the known ecology of these species, can inform us about how these five similar but independent lineages successfully came to occupy transcontinental ranges across northern North America to become integral members of forest communities. This combination of ecology and genetics is part of a growing examination of the interaction between evolutionary history and the ecological processes determining the makeup of assemblages and communities (*Ricklefs, 1987*; *Ricklefs, 2007*; *Johnson & Stinchcombe, 2007*; *Andrew et al., 2013*).

Because of the close evolutionary history, ecological similarity, and transcontinental distribution of these five species, we hypothesized that they might share similar historical patterns across northern North America. Many transcontinental vertebrate species and species complexes have a pattern of mtDNA genetic divergence across North America showing a split between a western coastal lineage and an eastern lineage (e.g., *Milot, Gibbs & Hobson, 2000*; *Omland et al., 2000*; *Arbogast & Kenagy, 2001*; *Kimura et al., 2002*; *Ruegg & Smith, 2002*; *Peters, Gretes & Omland, 2005*; *Milá, Smith & Wayne, 2007*). This pattern has been largely regarded as a result of Pleistocene glacial cycles and the accompanying climatic and ecological changes (*Pielou, 1991*; *Arbogast & Kenagy, 2001*; *Weir & Schluter, 2004*). Thus, we refine our hypothesis of shared histories to include an expectation of a clear genetic break between eastern and western phylogroups.

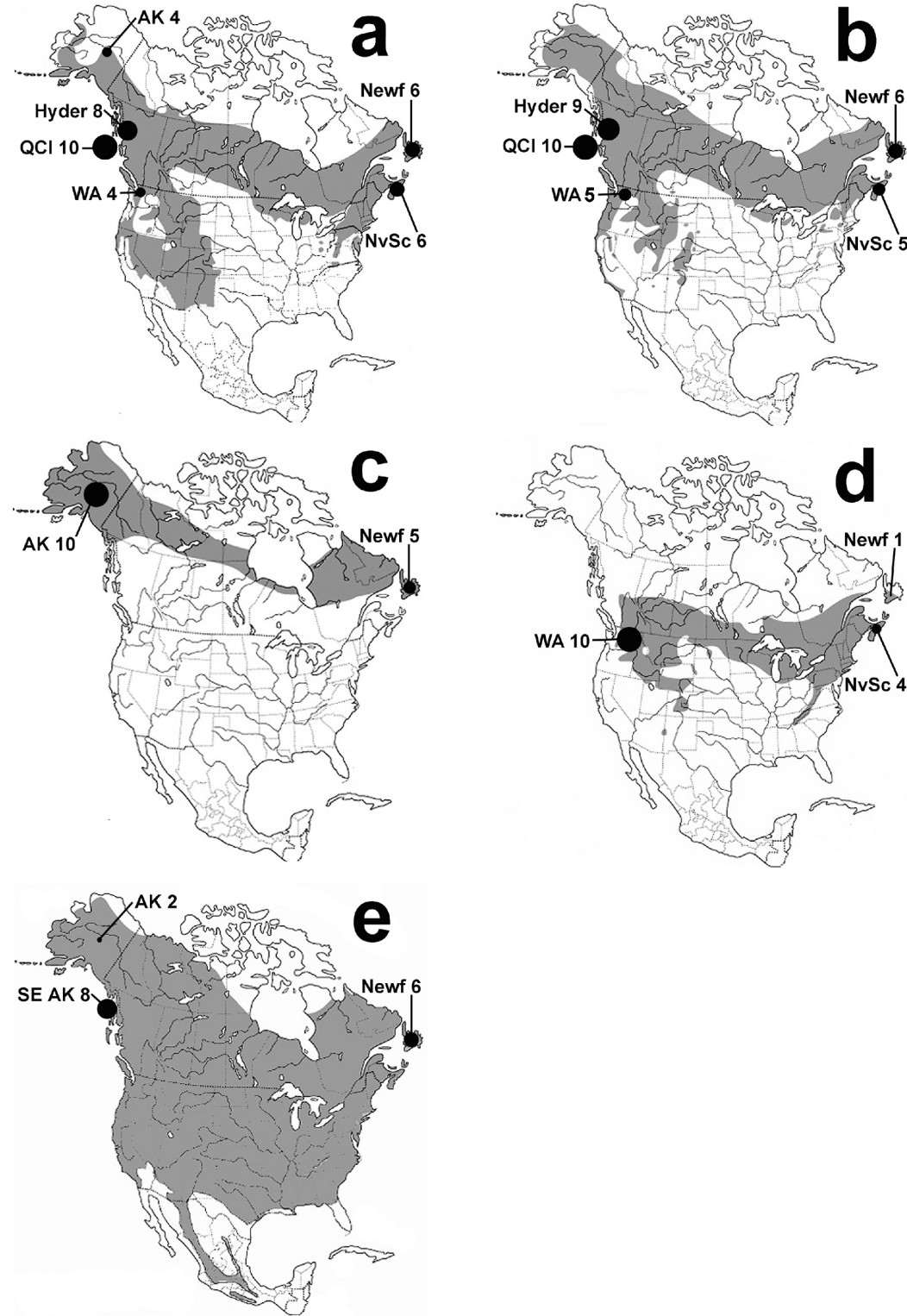

**Figure 1 Thrush distribution maps.** Maps of species breeding ranges with sample locations shown: Hermit Thrush (A), Swainson's Thrush (B), Gray-cheeked Thrush (C), Veery (D), and American Robin (E). Maps are based on the Birds (continued on next page...)

**Figure 1 (...continued)**
of North America Online http://bna.birds.cornell.edu/bna and *AOU (1998)*. Black circles are proportional in size to the number of individuals sampled from each location. Locations are eastern: Nova Scotia (NvSc) and Newfoundland (Newf); and western: Interior Alaska (AK), Southeast Alaska (SE AK), Hyder, Alaska (Hyder), Queen Charlotte Islands (QCI), and Washington state (WA).

We asked three questions: (1) Is there a pattern of genetic divergence between eastern and western populations, as suggested by other vertebrate studies? (2) Do coalescent events, such as lineage divergence, between eastern and western populations occur at similar times, and do these match historic glacial events? (3) Did these five ecologically similar, co-distributed thrush species come to occupy their ranges across northern North America in the same way, showing similar patterns of expansion?

## METHODS

### Sampling and mtDNA sequencing

The five migratory North American thrush species in this study represent all of the thrushes that are distributed across North America at higher latitudes, where they have mostly or partly overlapping breeding ranges (Fig. 1). We sampled thrush assemblages on each side of the continent to understand continental-scale patterns; finding finer-scale phenomena such as the location and shape of contact zones or clines between possible eastern versus western clades was not one of our goals, and we do not ask how these lineages came to exist on the continent itself (see, *Outlaw et al., 2003*; *Voelker et al., 2007*; *Voelker et al., 2009*). For comparisons across the continent of North America, we used two main sample regions along the northern coasts: Eastern = Nova Scotia (NvSc) and Newfoundland (Newf), Canada; and Western = interior Alaska (AK); southeast Alaska (SE AK); Hyder, Alaska (Hyder); Queen Charlotte Islands, Canada (QCI); and Washington state (WA; Fig. 1). Specimen voucher numbers and GenBank accession numbers are listed in Table A.1.

Total genomic DNA was extracted from muscle tissue following *Glenn (1997)* or DNeasy DNA purification kit protocols (Qiagen, Valencia, CA). Most or all of the cytochrome *b* gene was amplified using the reverse primer H16064 (*Harshman, 1996*) and the following forward primers: L14703 (C Huddleston, pers. comm., 1997) for Hermit Thrush and Gray-cheeked Thrush (1,143 bp); L14841 (*Kocher et al., 1989*) for Veery (1,045 bp); and L1650ND5 (*Winker & Pruett, 2006*) for Swainson's Thrush (1,094 bp) and American Robin (1,143 bp). All amplifications were performed using standard polymerase chain reaction (PCR) protocols (*Hillis, Moritz & Mable, 1996*) and cycle sequenced using Big Dye Terminator 3.1 and sequenced in both directions on an ABI 373, 3100, or 3130*xl* automated sequencer (Applied Biosystems Inc., Foster City, CA). We sequenced the mtDNA gene cytochrome *b* because it is a well-studied gene with a fairly constant rate of evolution and has proven useful in many phylogeographic and population genetic studies (*Moore & DeFilippis, 1997*; *Avise, 2000*).

## Summary statistics and haplotype networks

Mitochondrial sequence data were edited and checked for stop codons indicative of nonfunctional nuclear copies using Sequencher 4.7 (Gene Codes Corp., Ann Arbor, MI). Using DnaSP version 4.20.2 (www.ub.es/dnasp/; *Rozas et al., 2003*), sequences were examined for variable base pairs, haplotype variation (H), segregating sites (S), haplotype diversity (h), and nucleotide diversity per site ($\pi$). Statistical parsimony networks were made with TCS 1.21 (http://darwin.uvigo.es/software/tcs/html; *Clement, Posoda & Crandall, 2000*) to visualize haplotype relationships.

## Phylogenetic analyses

Based on preliminary analyses, we noted that a pattern of deep divergence within Hermit and Swainson's thrushes appeared to be similar to that occurring between the Gray-cheeked Thrush and Veery. Therefore, separate from our within-species analyses of eastern and western populations, we also conducted analyses to understand this possibly similar historic divergence event between species.

The nucleotide substitution model for each species was selected using PAUP* 4.0b10 (*Swofford, 2001*) and the Akaike Information Criterion (AIC) for model selection as implemented in Modeltest 3.6 (http://darwin.uvigo.es/software/modeltest.html; *Posada & Crandall, 1998*; *Posada & Buckley, 2004*). The best-fit maximum likelihood models were used in reconstructing phylogenetic trees for each species: HKY for American Robin and Gray-cheeked Thrush; TrN for the Veery and the combination of Gray-cheeked Thrush and Veery; TrN + I for the Hermit Thrush; and K81uf + I for Swainson's Thrush.

Phylogenetic trees for each species were reconstructed in MrBayes 3.1.2 (http://mrbayes.csit.fsu.edu; *Huelsenbeck & Ronquist, 2001*; *Ronquist & Huelsenbeck, 2003*; *Altekar et al., 2004*) and rooted with closely related outgroup taxa. Outgroup sequences were acquired from GenBank or from UA Museum specimens (Table A.2). Four independent runs starting from random trees were used for each species to ensure that the Markov chain converged on the optimal likelihood value. Trees were sampled every 10,000 generations, and the analyses were run for 8 million generations. All trees sampled before the Markov chain plateaued were discarded (the burnin), and remaining trees were used to approximate posterior probabilities for each phylogeny (*Huelsenbeck & Ronquist, 2001*). A burnin of 100,000 generations was sufficient in all species. The remaining 791 trees were then imported into PAUP* 4.0b10 (*Swofford, 2001*), where 50% majority rule consensus trees were generated with the posterior probabilities of each clade recorded as the percentage of that clade occurring among all the sampled trees (*Huelsenbeck & Ronquist, 2001*).

## Historic population changes

Changes in the site-frequency pattern of DNA polymorphisms that may be associated with past changes in population size were assessed using $R_2$ and Fu's $F_s$ statistics, as implemented in DnaSP 4.20.2 (*Rozas et al., 2003*; *Romis-Onsins & Rozas, 2002*). We chose $R_2$ and Fu's $F_s$ because they are more powerful tests than statistics based on

mismatch distributions, and $R_2$ provides superior estimates when sample sizes are small (*Romis-Onsins & Rozas, 2002*). The probability of our results under a model of constant population size was determined in DnaSP version 4.20.2 (*Rozas et al., 2003*) with 1,000 coalescent simulations based on observed $\theta (2N_e\mu)$ per gene, where $N_e$ is the effective population size and $\mu$ is the mutation rate per sequence per generation.

## Coalescent analyses

To estimate divergence times we used the coalescent program Isolation with Migration (IM version 10.10.07; http://lifesci.rutgers.edu/~heylab/HeylabSoftware.htm; *Hey & Nielsen, 2004*), which uses a Markov chain Monte Carlo (MCMC) approach. IM incorporates effective population sizes and migration rates while simultaneously estimating divergence time. Using IM, we estimated the time of divergence (*t*) between eastern and western populations and the time to most recent common ancestor (TMRCA). These parameters were scaled to the neutral mutation rate, making it possible to directly compare results between species. We compared *t*-values among the five species to examine coalescent patterns between eastern and western populations. To determine whether divergence dates *within* Hermit and Swainson's thrushes were similar to the divergence *between* the Gray-cheeked Thrush and the Veery we compared estimates of TMRCA.

To make a rough estimate of the timing of divergences, we converted *t*-values and TMRCA-values from IM to time in years using a generation time of one year and the estimated mutation rate of about 2% sequence divergence per million years for mtDNA in birds (*Hey & Nielsen, 2004*; *Lovette, 2004*; *Weir & Schluter, 2008*). This estimate is imprecise, but it enables us to roughly date these divergences (*Weir & Schluter, 2008*; *Ho et al., 2005*; *Ho et al., 2011*; *Pereira & Baker, 2006*).

At least three runs were performed in IM for each species: an initial run to estimate appropriate priors and then two additional independent runs with identical conditions but different random number seeds to confirm convergence. The runs with the highest effective samples sizes (ESS) were chosen to report results.

To set an upper prior for *t*, we assumed that the time since divergence could not be older than TMRCA, and we used the upper 95% credible interval value from preliminary runs to set the upper bound for *t* in each species (*Peters et al., 2007*). We ran IM for a different number of total steps for each dataset based on preliminary runs to ensure that the lowest ESS values were at least 500 (*Hey & Nielsen, 2004*): Hermit and Swainson's thrushes were run with 15,000,000 steps; the Gray-cheeked Thrush, Veery, and American Robin were run using 10,000,000 steps; Gray-cheeked Thrush and Veery combined as one dataset was run for 20,000,000 steps. For all species we used a burnin of 1,000,000 steps.

## Testing divergence hypotheses

To test the hypothesis of simultaneous divergence or establishment times across northern North America, we used msBayes (www.msbayes.sourceforge.net/; *Hickerson, Stahl & Lessios, 2006*; *Hickerson, Stahl & Takebayashi, 2007*). This program uses an approximate Bayesian computational (ABC) framework that tests for simultaneous divergence across multiple co-distributed taxon pairs (taxon pair = taxon with two populations) using

a hierarchical model that incorporates intrinsic variation such as ancestral coalescence and among-taxon demographic histories (*Hickerson, Stahl & Lessios, 2006*; *Hickerson, Stahl & Takebayashi, 2007*). This method allows for the simultaneous estimation of three hyperparameters that characterize the mean ($E[\tau]$), variability ($\Omega$), and number of separate divergence events ($\Psi$) across multiple population pairs. ABC obtains these estimates by simulating data and their summary statistics from the joint prior distribution under a model and then sampling from the resulting joint posterior distribution using probabilities based on the similarity between the summary statistic vector for the observed versus the simulated data (*Hickerson, Stahl & Lessios, 2006*; *Hickerson, Stahl & Takebayashi, 2007*). These methods are effective even with population sample sizes of five or less (*Hickerson, Stahl & Takebayashi, 2007*).

We examined two datasets with msBayes: (1) the five thrush taxa with eastern and western populations to test our main hypothesis; and (2), in a *post hoc* analysis given the mtDNA results, a three 'taxon' set of Hermit Thrush (E-W) and Swainson's Thrush (E-W) phylogroups and Gray-cheeked Thrush and Veery combined. The three 'taxon' set was based on the observation that the pattern of deep divergence within Hermit and Swainson's thrushes appeared to be similar to the divergence between the Gray-cheeked Thrush and Veery. This second analysis thus enabled us to test a secondary hypothesis that these three pairs of mtDNA clades might have a similar timing of divergence. The divergent clades within Hermit and Swainson's thrushes did not perfectly match eastern and western sampling locations (e.g., a few western birds had eastern haplotypes), so for this three 'taxon' analysis we used the phylogroups labeled eastern and western, based on sample locations. More details are given in Results.

We ran two million simulations in msBayes using the following starting parameters for the upper and lower bounds of prior distributions: $\theta$ lower = 0.5 (default), $\theta$ upper = 20.0 for the five-taxa dataset, and $\theta$ upper = 5.0 for the three 'taxon' dataset (based on the highest $\pi_W$ from observed summary statistics as recommended by *Hickerson, Stahl & Lessios (2006)*), $\tau$ upper = 10.0 for the five-taxa dataset, and $\tau$ upper = 15.0 for the three 'taxon' dataset (based on relatively recent divergence in the last 1 or 1.5 million years), migration rate upper = 10.0 (some migration is possible), recombination rate upper = 0.0 (mtDNA is unlikely to have recombination in birds), and ancestral population size upper = 0.5 (default). We report joint posterior estimates based on the summary statistic vector **D** that includes 13 summary statistics (see *Hickerson, Stahl & Lessios, 2006*) per taxon pair. We sampled the posterior distribution with a tolerance of 0.0005 and 0.00025, which yielded estimates based on 1000 and 500 draws from the joint posterior distribution, given that there were two million simulated draws from the joint prior. Results are presented using a tolerance of 0.00025, because this sampling parameter showed better resolution in the posterior probability density graph (peaks were more cleanly shaped), although results were very similar for both tolerance levels.

**Table 1 Table of genetic diversity and population size analyses.** Measures of genetic diversity and historic population size analyses calculated in DnaSP v.4.20.2 (*Rozas et al., 2003*) for each species total, eastern and western populations, and eastern and western phylogroups in the two species with deep divergences.

| Species | $n$ | $H$ | $S$ | $h$ | (SD) | $\pi$ (per site) | (SD) | $R_2$ | $P(R_2)$ | Fu's $F_s$ | $P$ (Fu's $F_s$) |
|---|---|---|---|---|---|---|---|---|---|---|---|
| Hermit Thrush | 38 | 19 | 37 | 0.93 | (±0.02) | 0.0118 | (±0.0005) | 0.18 | 0.977 | −0.06 | 0.556 |
| East | 12 | 5 | 4 | 0.73 | (±0.11) | 0.0009 | (±0.0002) | 0.13 | 0.258 | −1.82 | 0.050 |
| West | 26 | 14 | 33 | 0.81 | (±0.04) | 0.0105 | (±0.0013) | 0.18 | 0.943 | 0.48 | 0.594 |
| E. phylogroup | 21 | 11 | 11 | 0.90 | (±0.05) | 0.0019 | (±0.0003) | **0.08** | **0.032** | **−5.34** | **0.003** |
| W. phylogroup | 17 | 8 | 8 | 0.82 | (±0.08) | 0.0013 | (±0.0003) | **0.09** | **0.017** | **−3.73** | **0.003** |
| Swainson's Thrush | 35 | 24 | 42 | 0.92 | (±0.04) | 0.0088 | (±0.0005) | 0.10 | 0.367 | **−6.92** | **0.031** |
| East | 11 | 10 | 19 | 0.98 | (±0.05) | 0.0032 | (±0.0007) | **0.08** | **0.000** | **−6.12** | **0.002** |
| West | 24 | 15 | 32 | 0.84 | (±0.08) | 0.0063 | (±0.0015) | 0.09 | 0.154 | −3.03 | 0.115 |
| E. phylogroup | 16 | 14 | 24 | 0.98 | (±0.04) | 0.0032 | (±0.0006) | **0.05** | **0.000** | **10.21** | **0.000** |
| W. phylogroup | 19 | 10 | 12 | 0.74 | (±0.11) | 0.0014 | (±0.0004) | **0.06** | **0.000** | **−6.28** | **0.000** |
| Gray-cheeked Thrush | 15 | 11 | 12 | 0.94 | (±0.05) | 0.0018 | (±0.0003) | **0.07** | **0.000** | **−8.11** | **0.000** |
| East | 5 | 3 | 4 | 0.70 | (±0.22) | 0.0014 | (±0.0006) | 0.29 | 0.543 | 0.28 | 0.552 |
| West | 10 | 8 | 8 | 0.93 | (±0.08) | 0.0015 | (±0.0003) | **0.09** | **0.000** | **−5.63** | **0.000** |
| Veery | 15 | 5 | 5 | 0.56 | (±0.14) | 0.0008 | (±0.0003) | 0.12 | 0.126 | **−2.17** | **0.025** |
| East | 5 | 3 | 2 | 0.80 | (±0.16) | 0.0010 | (±0.0003) | 0.25 | 0.331 | −0.48 | 0.236 |
| West | 10 | 3 | 3 | 0.38 | (±0.18) | 0.0006 | (±0.0003) | 0.21 | 0.489 | −0.46 | 0.159 |
| American Robin | 16 | 5 | 6 | 0.68 | (±0.09) | 0.0013 | (±0.0003) | 0.12 | 0.105 | −0.37 | 0.417 |
| East | 6 | 2 | 1 | 0.33 | (±0.22) | 0.0003 | (±0.0002) | 0.37 | 1.000 | 0.00 | 0.534 |
| West | 10 | 4 | 5 | 0.71 | (±0.12) | 0.0016 | (±0.0004) | 0.17 | 0.296 | 0.44 | 0.606 |

**Notes.**

Measures of diversity are: $n$, sample number; $H$, number of haplotypes; $S$, segregating sites; $h$, haplotype diversity; $\pi$ (per site), nucleotide diversity.

$R_2$ and Fu's $F_s$ were used to measure historical population changes. 1,000 coalescent simulations were used to determine the probability of our results under a model of constant population size. Significant results are shown in bold ($P < 0.05$).

## RESULTS

### Genetic variation

The five thrush species had varying degrees of intraspecific genetic diversity, with the lowest number of haplotypes being 5 and the highest number 24 (Table 1). For all five species, more than 50% of the nucleotide substitutions were third position synonymous changes.

Two very different broad patterns were observed among species in the statistical parsimony networks (Fig. 2). Hermit and Swainson's thrushes had two deeply divergent lineages separated by 21 and 14 nucleotide differences, respectively; we will hereafter refer to these phylogroups as eastern and western (Fig. 2). The eastern phylogroups are primarily made up of individuals from eastern North America and interior Alaska (the latter has a stronger avifaunal affinity with eastern than western North America in these taxa; *Phillips, 1991*). The western group is primarily made up of individuals from southeast Alaska. Gray-cheeked Thrush, Veery, and American Robin, in contrast, showed no differences greater than two nucleotide substitutions between closest haplotypes (Fig. 2). However, each species had a slightly different pattern of relationship between

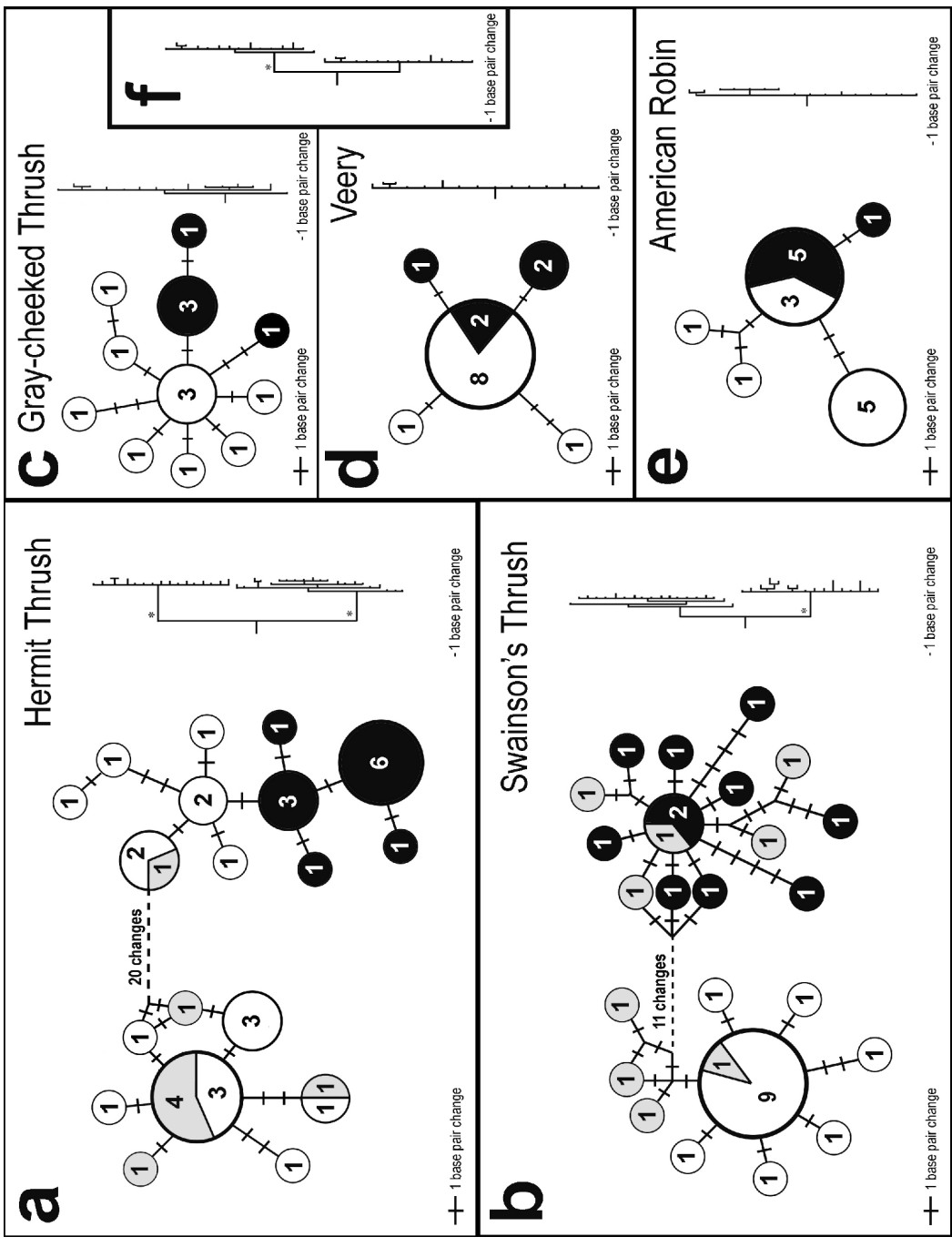

**Figure 2 Haplotype networks and trees.** Statistical parsimony networks showing haplotype relationships and the number of individuals with each haplotype. Shading indicates general sampling areas in North America; black, eastern; white, western; and gray, Hyder, Alaska. The size of each circle is proportional to the number of individuals with each haplotype. The length of connecting lines is proportional to the number of base pair differences between haplotypes. The phylograms on the right are sized proportionally to each other. Phylogeographically important nodes with Bayesian posterior probabilities of 1.0 are shown with an asterisk. Inset phylogram (F) shows the Gray-cheeked Thrush (top clade) and Veery (bottom clade) combined.

haplotypes sampled from eastern and western locations, and the two species with divergent lineages (Hermit and Swainson's thrushes) had haplotypes from both the eastern and western phylogroups in western populations (Fig. 2). In both species, individuals from Hyder, Alaska possessed haplotypes from both phylogroups, suggesting a zone of contact between eastern and western populations (Figs. 2A and 2B). The eastern Hermit Thrush phylogroup included one individual from Hyder and all of the Washington, interior Alaska, and eastern individuals, whereas the western phylogroup had the majority of the Hyder and all of the QCI individuals (Fig. 2A). The eastern Swainson's Thrush phylogroup contained five individuals from Hyder and all of the eastern individuals (Nova Scotia and Newfoundland), and the western phylogroup had all the QCI and Washington individuals and the remaining four Hyder birds (Fig. 2B).

## Phylogenetic patterns

The same two general patterns observed among the five species' haplotype networks (Fig. 2) were also observed in the Bayesian phylogenetic trees: Hermit and Swainson's thrushes had two divergent lineages with high posterior probabilities, and the other species had much less structure (Fig. 2). A Bayesian tree of the relationship between the Gray-cheeked Thrush and the Veery also had high posterior probabilities for nodes associated with the species-level split (Fig. 2F). The Bayesian phylogram of the Gray-cheeked Thrush and Veery combined showed an overall similarity to the within-species divergences observed in Hermit and Swainson's thrushes (Fig. 2F).

## Historical population changes

Significant excess of low frequency polymorphism, potentially associated with deviations from constant population size (a signal of recent rapid population expansion) were detected in: the combined Gray-cheeked Thrush sample locations and also in its samples from western North America, the eastern sampling locations of Swainson's Thrush, and both the eastern and western phylogroups of Hermit and Swainson's thrushes (Table 1). Only the Gray-cheeked Thrush had a significant signal of recent rapid population expansion for an entire species (Table 1). Veeries and American Robins did not differ significantly from a model of historic population stability (Table 1). The eastern population of American Robins, all Hermit Thrushes combined, and the western population of Hermit Thrushes had very high probabilities ($P > 0.94$) of observed $R_2$ values under an equilibrium model of historical population stability (Table 1).

## Coalescent analyses

Divergence times inferred using IM were also consistent with phylogenetic results, indicating two main patterns: deeper divergence within Hermit and Swainson's thrushes and between the Gray-cheeked Thrush and the Veery, and shallow divergence within the Gray-cheeked Thrush, Veery, and American Robin. IM analyses showed strongly unimodal posterior distributions for $t$ and TMRCA for all thrushes. Both the Hermit Thrush and American Robin divergences between eastern and western sampling locations had posterior distributions of $t$ with tails that did not approach zero, effectively making

**Table 2 IM values and divergence times.** IM values and divergence time converted to years for $t$ and TMRCA. Top rows are smoothed IM values and 95% credible intervals scaled to the neutral mutation rate of divergence time ($t$) between eastern and western samples in each species and the time to most recent common ancestor (TMRCA; divergence for clades). Bottom rows are IM values converted to time in years ($T$) assuming a generation time of one year and 2% sequence divergence per million years where the mutation rate ($\mu$) is $1 \times 10^{-8}$ substitutions/site/lineage/year and $T = Lt/\mu$, where $L$, the length of the sequence in base pairs. Numbers in bold represent the biologically important divergence measure for each species in describing continent-wide divergence.

| Species | $t$ | 95% low–high | TMRCA | 95% low–high |
|---|---|---|---|---|
| Hermit Thrush | 0.65 | 0.44–na* | **11.79** | **8.26–17.12** |
| | 56,430 | 38,373–na* | **1,031,601** | **722,240–1,498,023** |
| Swainson's Thrush | 0.72 | 0.39–1.67 | **8.23** | **5.44–12.50** |
| | 65,841 | 35,384–152,733 | **752,002** | **496,810–1,142,934** |
| Gray-cheeked Thrush | **0.74** | **0.31–2.48** | 1.37 | 0.79–3.12 |
| | **65,092** | **27,017–216,833** | 119,528 | 68,714–273,123 |
| Veery | **0.34** | **0.08–2.30** | 0.73 | 0.34–2.51 |
| | **32,651** | **7,291–220,450** | 69,971 | 32,536–240,191 |
| American Robin | **0.19** | **0.09–3.30** | 1.31 | 0.62–3.25 |
| | **16,807** | **8,180–na*** | 114,304 | 54,514–284,296 |
| Gray-cheeked Thrush and Veery | na | na | **5.99** | **5.11–9.66** |
| | | | **572,967** | **489,234–924,641** |

the upper 95% credible intervals infinity; however they also had clearly defined unimodal peaks. In both cases we used the upper 95% credible interval value estimated for TMRCA to set an upper bound on $t$ because we assumed $t$ could not be greater than TMRCA (Fig. 3). The Hermit Thrush posterior distribution for $t$ peaked over a range similar to other species' eastern-versus-western population $t$-values, and the distribution values that went to infinity were flat but very close to zero (not shown). This result for the Hermit Thrush suggests that the eastern population diverged recently, within the last 100,000 years before present (ybp), from the western population (all individuals sampled from the west of North America regardless of phylogroup). However, given other evidence that might reduce divergence estimates, such as the Hyder-region contact zone and likelihood of gene flow, we cannot rule out the possibility that this divergence occurred much earlier ($\leq 1.5$ million ybp; Table 2, Fig. 3) with subsequent secondary contact between lineages.

Credible intervals for TMRCA and $t$ broadly overlapped in the American Robin and Veery. These species also exhibited little genetic structure, and eastern and western individuals shared haplotypes, suggesting little or no divergence across the continent (Table 2, Figs. 2 and 3). Gray-cheeked Thrush populations had a TMRCA date with 95% credible interval that overlapped the credible intervals of $t$ (Fig. 3), and eastern and western populations did not share haplotypes. This indicates an older divergence event between eastern and western Gray-cheeked Thrushes than in the American Robin or the Veery (Figs. 2 and 3). Hermit and Swainson's thrushes showed deep divergences (TMRCA = 1.03 million ybp and 750,000 ybp, respectively) between eastern and western phylogroups, but

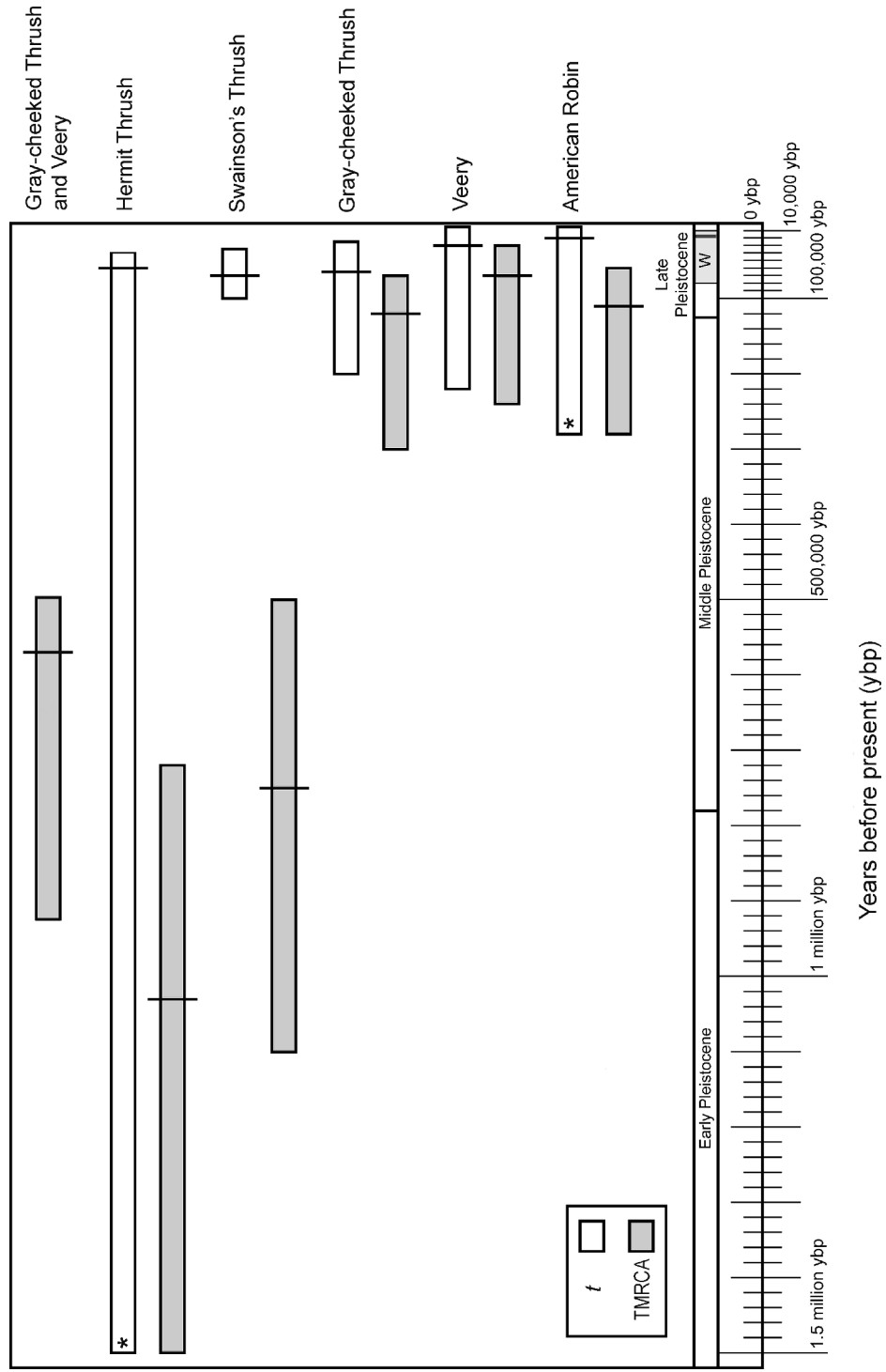

**Figure 3** **Divergence time estimates in graphic form.** Divergence time in years before present converted from smoothed IM values of divergence times with 95% credible intervals for *t* (white bars) and TMRCA (gray bars), assuming a one-year generation time and 2% 

**Figure 3 (...continued)**
sequence divergence per million years for cyt *b* in passerines. Along the bottom is a geological time scale showing different segments of the Pleistocene and the Wisconsin glacial period (*W*). The last glacial maximum (∼18,000 ybp) is shown with a thick line within the late Wisconsin. An asterisk indicates that the upper 95% credible interval value was used from TMRCA (see methods).

eastern phylogroup haplotypes were found in eastern and western populations, resulting in a shallow divergence ($t = 70,000$ ybp and 60,000 ybp, respectively) between the two sides of the continent (Table 2, Figs. 2 and 3). These results parallel results from our other analyses for these two species. The TMRCA 95% credible interval (490,000–925,000 ybp) between the Gray-cheeked Thrush and Veery overlapped the TMRCA 95% credible intervals for divergence between eastern and western phylogroups within the Hermit Thrush (720,000 ybp–1.5 million ybp) and within Swainson's Thrush (500,000 ybp–1.14 million ybp), which also parallels our other results (Table 2, Figs. 2 and 3).

### Testing divergence hypotheses

The results from msBayes showed that among the five thrush species there were as many as five different divergence events, while the 'three' taxon dataset of Hermit Thrush, Swainson's Thrush, and Gray-cheeked Thrush and Veery combined indicated a single shared divergence event.

For the five thrush species, the ratio of variance to mean divergence time was $\Omega = 2.15$ (95% quantiles $= 0.94$–6.56), which indicated multiple divergence events as estimated in msBayes ($\Omega = 0$ is expected for a set of population pairs with one divergence event). The number of divergence times across the five taxon pairs was five on the density graph (Fig. 4), with the highest point twice as high as all other values; however, there was a medium-density flat line across the other values that was slightly higher (around a mode of $\Psi = 2.36$; 95% quantiles $= 1.00$–5.00). This means that we can reject the hypothesis of one divergence event and that five is most likely, although there is a possibility that anywhere from two to five divergence events occurred. These results thus do not support a similar pattern of transcontinental occupancy of northern North America for the five thrush species.

Homogeneity in divergence time estimates for the three 'taxon' dataset of Hermit Thrush, Swainson's Thrush, and Gray-cheeked Thrush and Veery combined yielded a ratio of variance to mean divergence times of $\Omega = 0.00$ (95% quantiles $= 0.00$–2.56) and a value for the number of divergence times across taxon pairs of $\Psi = 1.02$ (95% quantiles $= 1.00$–2.86), which supports a history of simultaneous divergence among these three relatively deep splits: east versus west phylogroups for Hermit and Swainson's thrushes and between the Gray-cheeked Thrush and Veery (Figs. 2 and 4).

## DISCUSSION

Two primary patterns of transcontinental divergence were found among these five North American thrush species. Shallow levels of divergence were observed between eastern and western populations of the Gray-cheeked Thrush, Veery, and American Robin, whereas

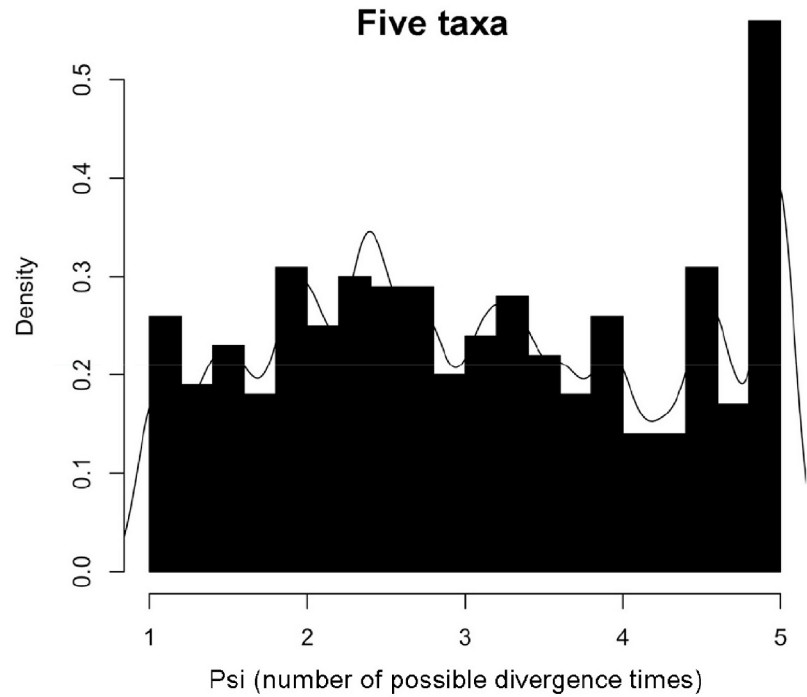

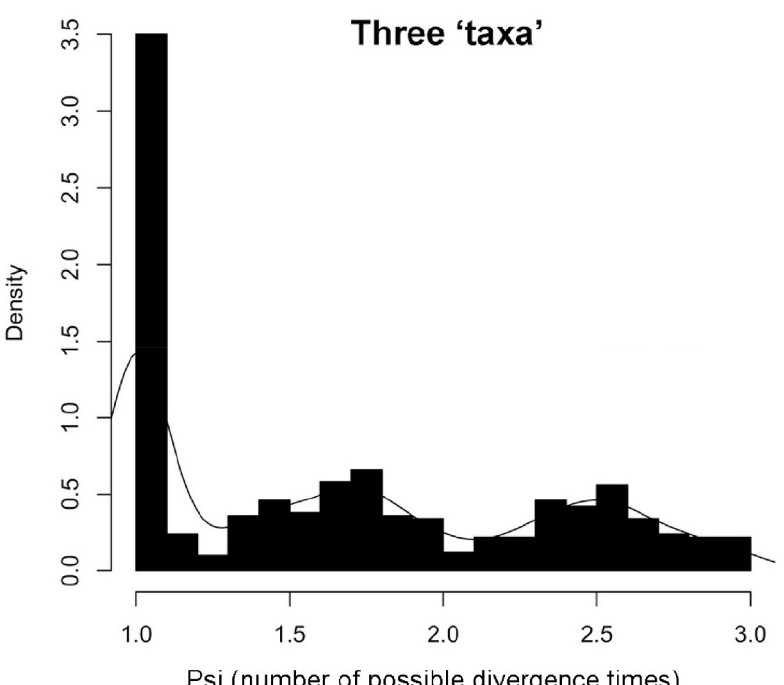

**Figure 4** **msBayes results.** msBayes posterior probability graphs of Ψ (Psi) for the five-taxon dataset (all five thrush species east–west populations) and the three 'taxon' dataset (Hermit Thrush and Swainson's Thrush phylogroups, and Gray-cheeked Thrushes and Veeries combined).

deeper divergences were seen in Hermit and Swainson's thrushes (Figs. 1 and 3). These five thrush species also appeared to have five significantly different coalescence events between eastern and western populations. On the other hand, the relatively deep divergences between eastern and western phylogroups *within* Hermit and Swainson's thrushes seemed to share a coalescence time with the split *between* Gray-cheeked Thrush and Veery (Fig. 2). These results indicate that despite their ecological similarity these five thrush species came to occupy northern North America in more than one way, with individual, species-level differences and two broad continental patterns.

## Occupancy across northern North America

We hypothesized that ecologically similar thrush species would share a pattern of transcontinental divergence, and that this pattern would be similar to continental divergences in mtDNA that have been found in other vertebrate species. However, the five thrush species did not share a single pattern of divergence, and only two of the three showed the type of transcontinental mtDNA split we had predicted based on other studies (*Milot, Gibbs & Hobson, 2000*; *Omland et al., 2000*; *Arbogast & Kenagy, 2001*; *Kimura et al., 2002*; *Peters, Gretes & Omland, 2005*; *Ruegg & Smith, 2002*). There is nothing obvious about the individual species' ecologies that seems concordant with these results.

The Veery and American Robin had little structure between eastern and western populations (Figs. 2D and 2E). The Gray-cheeked Thrush had no shared haplotypes between eastern and western populations, thus indicating a somewhat deeper divergence than in the Veery and Robin, but few mutations separated these populations (Fig. 2C). These three species may have spread across the continent to occupy their current ranges from single ancestral populations maintained through at least the last glacial maximum. Isolation by distance, gene flow, and extinction of a continental phylogroup may also affect the patterns observed in these three species.

Hermit and Swainson's thrushes had relatively deep divergences between eastern and western phylogroups (Figs. 2A and 2B). These splits did not exactly match the sampled eastern and western populations; however, this divergence is similar to patterns reported in other studies (*Arbogast & Kenagy, 2001*; *Weir & Schluter, 2004*). Previous population genetic research on Swainson's Thrushes found mtDNA sequence divergence between Pacific coastal and continental populations with a tension zone of secondary contact between them (*Ruegg & Smith, 2002*; *Ruegg, Hijmans & Moritz, 2006*; *Ruegg, 2008*). This indicates that eastern and western populations of Swainson's and Hermit thrushes were likely split during historic vicariant events and were isolated from one another throughout much of the Pleistocene. After the last glacial maximum (~19,000 ybp), they expanded across the continent into their current ranges and came into secondary contact (Fig. 3).

Because most avian species at higher latitudes are expected to have undergone postglacial population expansions, we expected to see relatively low values of $R_2$ and Fu's $F_s$ indicating these expansions (*Hewitt, 1996*). However, some populations in our study showed a signal of population stability (Table 1). This may be partly due to sampling error from small sample sizes (e.g., eastern population of the Gray-cheeked Thrush) and the

relatively deep splits within species and shared haplotypes in the western populations of Hermit and Swainson's Thrushes (Figs. 2A and 2B).

The eastern Swainson's Thrush population had a signal of recent rapid expansion, whereas the western population did not (Table 1). This is consistent with the findings of *Ruegg & Smith (2002)*, whose results showed expansion in continental populations of Swainson's Thrush but not in western coastal populations. This pattern has also been observed in other avian studies (*Milot, Gibbs & Hobson, 2000*; *Peters, Gretes & Omland, 2005*). However, at the phylogroup (rather than population) scale, our results had both eastern and western phylogroups of Swainson's Thrush showing strong signals of expansion, as did both of the Hermit Thrush phylogroups (Table 1). The eastern Swainson's Thrush and Hermit Thrush phylogroups did have lower negative Fu's $F_s$ statistics than the western phylogroups, supporting the possibility that the eastern phylogroups may have had greater expansion than the western ones (Table 1).

Gray-cheeked Thrush as a species had a signal of expansion (Table 1), as would be expected by a spread across northern North America from a common ancestral population after glacial recession from the last glacial maximum (Fig. 1; *QEN, 1997*). The separation of eastern and western haplotypes (Fig. 2) may indicate a relatively short vicariant split between populations of this species during, for example, the last glacial maximum, but it might also represent isolation by distance with limited transcontinental gene flow. The American Robin and Veery had no signals of population expansion. As the southernmost breeding members of this assemblage, it is possible that populations of these species did not expand significantly following the last glacial maximum.

## Patterns shared with other vertebrates

Several transcontinentally distributed North American bird and mammal taxa exhibit mitochondrial lineage breaks between the northwest coast and lineages found in the rest of their North American range (Table 3). These western-versus-eastern patterns of differentiation are widely considered to be the result of glacial history and corresponding climatic and ecological changes over time (*Pielou, 1991*; *Arbogast & Kenagy, 2001*; *Weir & Schluter, 2004*; *Ruegg, Hijmans & Moritz, 2006*). Although two of the east–west divergences found among these studies (Table 3) may pre-date the Pleistocene (*Omland et al., 2000*; *Toews & Irwin, 2008*), the divergence levels that we found among thrushes (Table 2, Fig. 3) are generally contemporaneous with these many Pleistocene-era divergences.

Our estimate of divergence time (TMRCA) between the eastern and western phylogroups in Swainson's Thrush is one or two orders of magnitude older than that estimated by *Ruegg & Smith (2002)*, who estimated the time of divergence between these two groups as 10,000 ybp. Differences in time estimates could be due to using different genes and different estimates of mutation rate. *Ruegg & Smith (2002)* used the mitochondrial control region and an estimated divergence rate of 14.8% per million years, which is a much higher mutation rate than is usually assumed for passerines (*Marshall & Baker, 1997*; *Bensch, Andersson & Akesson, 1999*; *Griswold & Baker, 2002*; *Bulgin et al., 2003*; *Perez-Tris et al., 2004*; *Davis et al., 2006*).

**Table 3 Other taxa with mtDNA breaks.** Other transcontinentally distributed vertebrates with mtDNA lineage breaks between the northwest coast and lineages found in the rest of their North American range.

| Species | mtDNA marker | Divergence estimate | | |
|---|---|---|---|---|
| | | % divergence | ybp | Source |
| **Birds** | | | | |
| Wood Duck (*Aix sponsa*) | control region | – | 124,000 − 10,000 | *Peters, Gretes & Omland (2005)* |
| Hairy Woodpecker (*Picoides villosus*) | NADH dehydrogenase subunit 2 | ~1.5% | 925,000 − 470,000 | *Klicka et al. (2011)* |
| Gray Jay (*Perisoreus canadensis*) | NADH dehydrogenase subunit 2 | 4.6–5.1% | 5,520,000 − 890,000 | *van Els, Cicero & Klicka (2012)* |
| Common Raven (*Corvus corax*) | cytochrome *b*, control region | 4.0–5.0% | ~2,000,000 | *Omland et al. (2000)* |
| Boreal Chickadee (*Poecile hudsonicus*) | control region, ATPase | – | 133,300 − 26,700 | *Lait & Burg (2013)* |
| White-breasted Nuthatch (*Sitta carolinensis*) | NADH dehydrogenase subunit 2 | 3.6–4.7% | 1,600,000 − 630,000 | *Spellman & Klicka (2007)* |
| Brown Creeper (*Certhia americana*) | NADH dehydrogenase subunit 2 | 1.5–3.0% | 2,500,000 | *Manthey, Klicka & Spellman (2011)* |
| Winter & Pacific wrens (*Troglodytes hiemalis* & *pacificus*) | NADH dehydrogenase subunit 2 | 6.24% | 4,300,000 | *Toews & Irwin (2008)* |
| Yellow-rumped Warbler (*Setophaga coronata*) | control region, ATPase 6 & 8 | 0.13–0.15% | 12,000 − 10,000 | *Milá, Smith & Wayne (2007)* |
| Yellow Warbler (*Setophaga petechia*) | control region | – | 100,000 − 18,500 | *Milot, Gibbs & Hobson (2000)* |
| Wilson's Warbler (*Cardellina pusilla*) | control region | – | 62,500 − 33,654 | *Kimura et al. (2002)* |
| Fox Sparrow (*Passerella iliaca*) | cytochrome *b*, NADH dehydrogenase subunit 2 | 1.47–1.91% | 770,000 | *Weir & Schluter (2004)* |
| **Mammals** | | | | |
| shrews (*Sorex bairdi* and *S. monticolus*) | cytochrome *b* | 4.7–5.7% | – | *Demboski, Stone & Cook (1999)* |
| northern flying squirrel (*Glaucomys sabrinus*) | cytochrome *b* | 4.3% | 1,200,000 − 770,000 | *Arbogast (1999)* |
| tree squirrels (*Tamiasciurus douglasii* + *mearnsi* and *T. hudsonicus*) | cytochrome *b* | 1.0–2.4% | 240,000 − 80,000 | *Arbogast, Browne & Weigl (2001)* |
| red-backed voles (*Myodes gapperi*) | cytochrome *b* | 3.3% | – | *Runck & Cook (2005)* |
| black bear (*Ursus americanus*) | control region, cytochrome *b* | 5%, 3.1–3.6 % | 1,800,000 | *Wooding & Ward (1997)*; *Stone & Cook (2000)* |
| American pine marten (*Martes americana*) | cytochrome *b* | 2.5–2.8% | – | *Demboski, Stone & Cook (1999)* |

## Divergences among thrushes

When we compared eastern and western population divergences among these five thrush species, the divergence dates (*t*) all occurred within the last 300,000 years (Table 2, Fig. 3).

However, our analysis of these five species indicated that there was more than one and perhaps as many as five different divergence or vicariance events for the five species (Table 1, Fig. 3). It thus seems quite possible (Fig. 4) that each species had its own unique history of how it came to occupy a transcontinental range in northern North America, even though they inhabit similar and in some cases nearly identical northern communities.

At a deeper level, examining the relatively deep events of cladogenesis in our datasets, the eastern and western phylogroup splits within Hermit and Swainson's Thrushes and the divergence between the Gray-cheeked Thrush and Veery occurred during a similar time interval (Table 2, Fig. 3). This suggests a shared divergence period within Hermit and Swainson's Thrushes and between the Gray-cheeked Thrush and the Veery and that the time that a lineage has had to occupy transcontinental North America affects whether it exhibits relatively deep east–west splits. Hermit and Swainson's Thrushes have existed as species much longer than the Gray-cheeked Thrush, Veery, and American Robin (∼4.0 and 2.6 million ybp versus ∼0.4, 0.4, and 0.32 million ybp, respectively; *Outlaw et al., 2003*; *Voelker et al., 2007*). Variation in the time available to spread across the continent could influence among-species phylogeographic heterogeneity, and the corroboration here between lineage age and higher-order pattern suggests that it does influence our results.

It is now generally agreed that glacial cycling in the Pleistocene created much of the observed interspecific and sister-species level divergences in many songbird species, especially in the northern hemisphere (*Klicka & Zink, 1997*; *Avise & Walker, 1998*; *Johnson & Cicero, 2004*; *Weir & Schluter, 2004*; *Lovette, 2005*). Our results indicate that the divergence events apparent in some thrush species also probably occurred within the Pleistocene (Table 2, Fig. 3). Paleoecological data suggest that forest habitat may have been present in North America to the east and west, just south of the last glacial maximum's southern expanse, while the center of the continent was grassland and desert (*Pielou, 1991*; *Crowley, 1995*; *QEN, 1997*). For forest-dependent species this could have been a significant barrier to gene flow, even among long-distance seasonal migrants, and may have caused relatively isolated breeding populations in forest refugia. When the glaciers receded, breeding ranges could expand into their current ranges (*Pielou, 1991*; *Weir & Schluter, 2004*; *Ruegg, Hijmans & Moritz, 2006*). This description of the last glacial maximum and expansion into new ranges may describe recently diverged species or populations. However, older glacial cycles in the Pliocene and the early- to mid-Pleistocene may have affected populations and species in similar ways, and thus created patterns such as those seen in the deeply divergent phylogroups within Hermit and Swainson's thrushes and between the Gray-cheeked Thrush and Veery.

In the broader context, patterns other than east–west divergences emerged during the Pleistocene. For example, the divergence between Gray-Cheeked Thrush and Veery is largely north–south (Fig. 1), as it is also between the American Robin and its sister the Rufous-collared Robin (*Turdus rufitorques*; *Voelker et al., 2007*). However, it seems that the majority of divergence patterns are east–west in northern North America. In addition, at this larger scale it is possible that the lack of transcontinental structure in the American Robin might be due to a rapid northern expansion into a region without other *Turdus*

species as competitors. This leaves open the question, however, of why similar expansions did not occur in the Wood Thrush and Varied Thrush, both of which lack congeners in North America. Thus, the role that competition might play in affecting these results is not clear.

*Weir & Schluter (2004)* found that many bird species complexes in boreal regions diverged into east (taiga) and west (Pacific coast) phylogroups about 1.2 ($\pm$0.10) million ybp. Because ice sheets did not form a single ice mass until the second half of the Pleistocene, it is likely that long periods of boreal fragmentation into eastern and western regions occurred during the early- to mid-Pleistocene (1.8–0.8 million ybp) when glaciers began to increase (*Barendregt & Irving, 1998*; *Weir & Schluter, 2004*). The 95% credible interval for estimated dates of divergence (TMRCA) between phylogroups in Hermit and Swainson's thrushes and between the Gray-cheeked Thrush and Veery overlap this period. Contemporary gene flow between Hermit and Swainson's thrush phylogroups (e.g., in Hyder, Alaska) and their relative lack of phenotypic differentiation as opposed to the Gray-cheeked Thrush and Veery, suggest that these two lineages within each species did not sufficiently differentiate during previous separation to achieve reproductive isolation and full biological speciation.

## CONCLUSION

From the perspective of community genetics, these five North American thrushes became widespread members of northern forests and woodlands in different ways. Multiple factors, from the local, ecological level (e.g., competition), to regional, evolutionary levels (e.g., climatic and glacial changes), were likely involved in producing the current transcontinental ranges observed in these five ecologically similar North American thrushes. It is of interest that there are species-level patterns but that two overriding patterns are also evident — i.e., a lack of homogeneity at one organizational level (five separate divergence levels within species is most probable) with evidence of concordance around two general patterns at another, among-species level, likely related to lineage age. This suggests that the processes that brought about these present continental assemblages are neither fixed, causing all species to have the same historical pattern, nor completely stochastic, in that there are two general patterns of divergence.

## ACKNOWLEDGEMENTS

Genetic samples were supplied by the University of Alaska Museum and the University of Washington Burke Museum. Thanks to C Barger, RW Dickerman, DD Gibson, AB Johnson, and TM Braile for their assistance in field and laboratory. KW thanks Canadian authorities for collecting permits. Thanks also to JL Peters and T Roberts for consultation, to MJ Hickerson and N Takebayashi for assistance and guidance on using msBayes, and to Martin Paeckert and an anonymous reviewer for comments on a previous draft.

# Appendix A

**Table A.1** **Voucher numbers and GenBank accessions.**

| Species | Museum[a] | Catalog numbers | GenBank accession |
|---|---|---|---|
| *Catharus guttatus* | UAM | 7322, 7564, 9989–93, 9995–8, 10108, 13235, 13415, 14351, 17601, 19819, 19821–2, 19824–6, 20779, 24436–8, 24440–2, 24444–8. | EU619718–EU619755 |
| | UWBM | 43131, 62639–40, 74551. | |
| *Catharus ustulatus* | UAM | 7323, 7523, 7525, 7538, 7540, 7570, 9978-85, 13411, 19829–42, 19844. | EU619756–EU619790 |
| | UWBM | 43114, 74064, G. K. Davis 220, G. S. Bergsma 40, W. C. Webb 14. | |
| *Catharus minimus* | UAM | 7440, 7457–8, 7596, 8965, 12984, 13208, 13405–7, 13410, 14546, 14669, 19812, 19814. | EU619791–EU619805 |
| *Catharus fuscescens* | UAM | 13414, 13416, 19845–7. | EU619806–EU619820 |
| | UWBM | 62067–8, 62071, 62073, 62078, 62083–4, 62136, 62144, 62151. | |
| *Turdus migratorius* | UAM | 7232–3, 14912–3, 13466, 13951, 14128, 14825, 14889, 14938, 24415–20. | EU619821–EU619836 |

**Notes.**

[a] University of Alaska Museum; University of Washington Burke Museum.

**Table A.2** **GenBank accessions for Outgroup taxa.**

| Species | Outgroup | GenBank accession |
|---|---|---|
| *Catharus guttatus* | *Catharus gracilirostris* | AY049497 |
| | *Catharus occidentalis* | AY049506 |
| *Catharus ustulatus* | *Catharus gracilirostris* | AY049497 |
| | *Catharus occidentalis* | AY049506 |
| *Catharus minimus* | *Catharus bicknelli* | AY049490 |
| | *Catharus fuscescens* | AY049495 |
| *Catharus fuscescens* | *Catharus bicknelli* | AY049490 |
| | *Catharus minimus* | AY049503 |
| *Turdus migratorius* | *Turdus libonyanus* | AY752389 |
| | *Turdus obscurus* | AY049484 |
| *C. minimus & C. fuscescens* | *Catharus frantzii* | AY049493 |
| | *Catharus occidentalis* | AY049506 |

### Funding

This project was supported by the University of Alaska Museum and an anonymous donor. The funders had no role in study design, data collection and analysis, decision to publish, or preparation of the manuscript.

### Grant Disclosures

The following grant information was disclosed by the authors:
University of Alaska Museum.

### Competing Interests

The authors declare that they have no competing interests.

### Author Contributions

- Carrie M. Topp and Kevin Winker conceived and designed the experiments, performed the experiments, analyzed the data, wrote the paper.
- Christin L. Pruett and Kevin G. McCracken performed the experiments, analyzed the data, wrote the paper.

### DNA Deposition

The following information was supplied regarding the deposition of DNA sequences:
GenBank: Voucher numbers for the specimens we used are provided in Table A.1.

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
