# Peer review of "How migratory thrushes conquered northern North America: a comparative phylogeography approach"

_PeerJ, doi:10.7717/peerj.206_

## Round 0.1 · original submission · Major Revisions

Dear authors
Thank you for submitting your manuscript to our journal. As you see our reviewers suggest a revision of your ms. If you are willing to do so, we would be happy to reconsider your revised manuscript.

Michael Wink

·

Basic reporting

The paper is well written, the structure is according to the common standard. All figures and tables are relevant and self-explanatory.

Experimental design

The overall desing of the study is appropriate. My only concern here is, that the basic aim of conducting a community genetics approach is not met by the study design. I have commented in detail why and what is the alternative. Finally, I think the terme "community genetics approach" must be definitely replaced in the paper title - e.g. by "comparative phylogeographic approach" (see comments to the authors).

Validity of the findings

All methods are sound and up to current standard. The sampling is appropriate for all analyses performed. The statistical analyses are sound and the results are thus robust and reliable.

Additional comments

The study by Winker et al. compares phylogeographic patterns of five North American thrush species with similar and mostly overlapping ecological niches. The main focus lies on the detection of an hypothesized intraspecific East-West divide which has been previously confirmed for many North American passerines. The data set is robust, the samplings are ok though could be more extensive for a phylogeographic approach in some of the study species. But in general, the sampling is appropriate for the detection of intraspecific splits and the molecular dating approach (also up to current standard). The paper compiles robust results based on sound data and some interesting aspects on the radiation of thrushes on the North American continent and I have enjoyed reading it. I would like to recommend it for publication after a minor revision.

I have a few minor concerns that should be considered prior to acceptance.

1. My main recommendation to the authors is to seriously reconsider whether this work is really based on a “community genetics approach”! To my understanding community ecology and particularly community genetics aim at describing and comparing diversity patterns of species assemblages in a much more comprehensive way than it is done here – honestly, an assemblage of five closely related species is far from being a comprehensive approach in the sense of community genetics, and I think the authors are aiming much too high here. Community genetics deals with some standard parameters of alpha diversity (when referred to local assemblages) such as species richness, phylogenetic diversity, functional diversity that cannot be assessed with the five-taxa data set in this study (in the sense of community genetics) neither are different assemblages compared in this study. Furthermore, in the strict sense not all five study species belong to the same assemblage (again in the sense of community ecology that usually deals with sets of local assemblages). The gray-cheeked thrush and the veery are largely allopatric at best parapatric at the Northeast coast (which is an important issue when interpreting the phylogeographic patterns). Thus in fact we are dealing with a northern assemblage (Fig. 1 a, b, c, e – excluding d because it has no overlap with c!) and a southern assemblage (Fig. 1 a, b, d, e – excluding c because it has no overlap with d!). A community genetics approach then would possibly compare local assemblages at sampling points WA with those at AK (Fig. 1), however with only five (locally four!) species this approach does not really make sense.
I strongly suggest that the authors reconsider their main aim of the study for what it in fact is: A comparative phylogeographic approach of closely related ecologically similar thrush species. In that context, this is a strong study based on clear hypotheses and expectations and provides reliable and robust results.

2. I was first not fully convinced by the three-taxon phylogroup classification as explained in l. 179 but considering the very close phylogenetic relationship of GCT and V I think a separate alternative analysis of the four Catharus species can be justified. Nevertheless, I appreciate the authors´ decision to lead the discussion largely based on the results of the five-taxa set that suggested more than one in favor of more than a single vicariance event – to my impression this is the scenario that matches the results better and I would like to encourage the authors to stick to that interpretation.

3. Patterns in other species: The authors made a strong effort in compiling examples from other bird and vertebrate species that show similar phylogeographic patterns to those found in their study species – including comparisons of separation time estimates. The whole paragraph l. 362 – 404 is important and valuable, however it has many direct citations of time estimates which make it a bit long. I would really like to see these comparisons in a table for target species, other passerines/birds and other vertebrates. To my personal feeling that would help the reader to get the “big picture” of Pleistocene speciation in the Nearctic at a glance. A few more recent papers on East-West divides in Nearctic passerines could be added here, e.g. Poecile hudsonicus (Lait & Burg 2011, Heredity), Poecile gambeli (Spellman et al. 2007, Mol Ecol 16), Sitta carolinensis (Spellman & Klicka 2007, Mol Ecol 16).

4. Patterns/ vicariance: It is good to have the study focused on East-West divides in the Nearctic, however at some points of the discussion I feel it could be helpful to shed a light on other patterns that directly correlate with the East-West vicariance events. For example, it is ok that the authors excluded C. bicknelli because of its restricted distribution at the eastern margin of the focal study region. However, the branching pattern of the Catharus tree in Outlaw et al. (2003) confirms a very close relationship of minimus, fuscescens and bicknelli and the short branch lengths of their terminal clades suggest rather simultaneous and very recent lineage splits among these three dating back to a single vicariance event. Unlike the East-West divides, this event rather shaped a North-South divide between C. minimus and C. fuscescens (with C. bicknelli possibly restricted to a marginal coastal refuge and post-Pleistocene secondary contact and overlap with C. fuscescens). I think this is a relevant issue that should be kept in mind and that should be discussed. Likewise, C. guttatus and its closest relative C. occidentalis represent a (however considerably older) Nearctic North-South divide and I think it is part of the “big picture” to briefly mention that there are also different phylogeographic patterns in the same study genus than the main focus East-West pattern.

A closer look on T. migratorius could be interesting, too. Apparently, there was no diversification in this species, it might have distributed from a single refuge in a rapid range expansion, but unlike in the four Catharus species across the whole continent all over North America (l. 322). This might be related to the biogeographic and phylogenetic origin of that species in which it differs from the other study species. The American robin is the only member of Turdus in the Nearctic and it is likely to have originated from a Pleistocene Central American radiation (Nylander et al. 2008). So by the time the American robin colonized the Nearctic there were no close relatives from genus Turdus competing for potential free niche space – unlike in Catharus species. That might be a possible explanation for the different patterns.

Reviewer 2 ·

Basic reporting

Pros: this paper analyzes co-distributed species that the authors say are both ecologically similar and different, and the patterns are possibly congruent in space but possibly not in time.

Experimental design

Cons:
Testing the same data twice (3 and 5 species analyses)
Not obvious why Bicknell’s excluded
One locus (mtDNA notoriously older splits)
Too few sample sites – if there are clines, the analysis is flawed
“Splits” between robins, veery, and gray-cheeked not splits at all. Splits between swainson’s clearly likely influenced by distance.

Validity of the findings

Cons:
Robins are not forest species, the others are
Not obvious why Bicknell’s excluded
This is not a “combination of ecology and genetics…” – there is no ecological data in the paper
Nothing in the paper suggests anything about how species colonized North America, only possibly that splits were not contemporaneous.

Calibration is an issue
Suggesting that a peak at “0” means one split, including incompletely sorted taxa

No mention of obvious taxonomic inconsistency (some splits species, others phylogroups).

Additional comments

Discussion too long by 3X – highly repetitive, lots of tangential material

---

## Round 0.2 · accepted · Accept

Dear authors
Thank you for resubmitting your manuscript to our journal. We are happy to accept your ms by now. Thank you for submitting your research results to PeerJ

Michael Wink